# An Experiment on the Psychological and Physiological Effects of Skin Moisturization on Lower Legs—In Expectation of Application to Nursing Practice at Hospitals

**DOI:** 10.3390/bs8100091

**Published:** 2018-09-30

**Authors:** Taichi Hitomi, Chigusa Theresa Yachi, Hajime Yamaguchi

**Affiliations:** 1The Occupational Therapy Department, Faculty of Health Science Technology, 1196 Kamekubo, Bunkyo Gakuin University, Fujimino-shi, Saitama 356-8533, Japan; 2School of International Humanities and Social Science, J.F. Oberlin University, Tokyo 194-0294, Japan; info@i-mental-fitness.co.jp (C.T.Y.); y-hajime@obirin.ac.jp (H.Y.)

**Keywords:** touch, skin, body and mind

## Abstract

This study hypothesized that moisturizing treatment of the skin has a positive effect on psychological and physiological aspects. In this experiment, the effect of touch with moisturizer for two minutes on the lower legs was measured in terms of brain activity, heart rate, and center of gravity unrest (X axis) in 10 healthy male and female experiment participants. The Right Laterality Ratio Score decreased after treatment, suggesting a relaxation effect of the treatment. Although it was not statistically significant, a decrease was observed. Heart rate decreased after the treatment at a level of statistical significance (*p* < 0.01), suggesting a relaxation effect of the treatment. Center of gravity unrest (X axis) increased after the treatment with statistical significance (*p* < 0.05). Therefore, skin moisturizing treatment was found to be effective both psychologically and physiologically in this experiment. The finding is expected to be applied to the field of nursing to support elderly people to enhance their mental well-being and balancing ability.

## 1. Introduction

It has been reported in various preceding studies that touching has positive psychological effects [1,2]. Yamaguchi [3] reported that when nurses touched patients, there was a possibility to enhance the natural healing power of the patients, and nurses were able to understand the patients’ anxiety and suffering. However, it was also reported that nurses have less chance to touch patients [4]. It is recognized that touching done by nurses has positive physical and psychological effects on patients. However, it is difficult for nurses to have enough time to touch patients in their daily practice.

In the nursing practice for elderly people, there are many patients whose physiological function of the skin is compromised due to aging. It has been recognized as very important for nurses to maintain hygiene and an appropriate level of moisture in the skin of elderly patients [5]. Yamamoto and Hayashi [6] reported that 50% of the time that nurses spent applying moisturizer was dedicated to the lower legs. The result of the study indicated that it was common practice for nurses to apply moisturizer to the lower legs of elderly patients. If the nurses could apply effective touching practice while applying the moisturizer, it would be expected that such practice could positively affect the patients. Nurses have chances to touch patients on various occasions in their nursing practices. In this research, both the physiological and psychological effects of application of moisturizer to the skin of the lower legs was examined.

The effect of skin moisturizing has already been reported, examining cuticle moisture level as an index [5,6,7]. However, in these studies, moisturizer was applied either to prevent skin problems, or to treat deteriorated skin conditions. There was no preceding study done examining the psychological effect of the application of moisturizer. In view of nursing practice, it is presumed that application of skin moisturizer would have two elements; one is touching, and the other is skin moisturizing.

Konno and Yoshikawa [8] used an index of firmness of feet touching the ground to examine psychological state and posture. Konno and Yoshikawa reported that the “Dohsa method”, a modality utilizing the touching enhanced relaxation level of the subjects, increased the subjective feeling of firmness of feet touching the ground, and eventually the psychological state and posture of the subjects was improved.

If application of moisturizer to the lower legs of patients was found to positively affect their psychological state, it would help improve nursing practice. This research examined the relaxation effect of moisturizer application to the lower legs, which already exists in regular nursing practices.

## 2. Materials and Methods

In this research, the experiment participants were healthy staff members who were working in a hospital. It was expected that elderly people would have various psychological and physical conditions, and there would be significant challenges to control the experimental conditions if elderly people were chosen as experiment participants. Since this research was considered basic research, healthy participants were recruited so that the experimental conditions could be strictly controlled. A written document was shown to the recruited experiment participants, which described that participation in the experiment was at their discretion and they would not be penalized for declining to participate. In addition to that, the content was verbally explained to them. The schema of the experiment was approved by the J. F. Oberlin University Research Ethics Committee (Approval No. 16044).

The therapist was a 37-year-old male, a certified psychologist, and a certified occupational therapist who was a staff member at the rehabilitation hospital. The therapist was never trained in any touching modalities.

The experiment room was approximately 100 m^2^, and it was used for rehabilitation activities in the hospital. The room temperature was kept at 25 to 27 °C. People were asked to stay away from this room during the experiment so that nobody was either coming in or out of the room. The room was kept quiet. The experiment was conducted from September to October in 2017.

To examine relaxation effect of moisturizer application to the lower legs, both a physiological and a body index were utilized. A HOT-1000 portable brain activity measurement device (Hitachi, 2-2 Kandaji-cho, Chiyoda-ku, Tokyo Niigakura Building, Japan) was used for the physiological index. This is used to monitor blood flow changes related to brain activity. When the part of the brain that corresponds to the path of the light is activated, blood flow and light absorption increases, and the amount of light returning to the detector decreases. The rate that the light decreases when returning to the detector is the principle behind how the HOT-1000 measures brain activity. The HOT-1000 uses light with a wavelength of around about 800 nm that is easily absorbed by the hemoglobin in the blood. Light scattered from above the scalp will diffuse and return to the detector. In the case where there is a good amount of hemoglobin in the region, much of the diffused light is absorbed. The HOT-1000 is powered with either AAA batteries or via USB, and Bluetooth connectivity makes this unit completely wireless. Moreover, since there are no cables attached, user feel no restrictions in movement or behavior, improving the nature of experimental designs. A Nintendo Wii Balance Board (Nintendo, 11-1 Shimobori Hachikate Town, Minami-ku Kyoto City, Japan) was used for the body index. To support the objective indices, the participants filled out a subjective questionnaire.

### 2.1. Physiological Index

The HOT-1000 device collected 10 data points per second of brain activity on both sides of the prefrontal area and heart rate. The Right Laterality Ratio Score (RLS) is considered useful to determine the difference in brain activity between the left and right prefrontal areas [9]. In this research, RLS was used as an index of relaxation effect. When the data showed a positive quantity, it meant that the brain activity of the right prefrontal area was more dominant than that of the left. When the data showed a negative quantity, it meant that the brain activity of the right prefrontal area was less dominant than that of the left. It was interpreted that the sympathetic nerve system was more active and the parasympathetic nerve system was suppressed when RLS was higher, and the parasympathetic nerve system was more active and the subject was more relaxed when RLS was lower [10]. RLS was calculated as follows.

RLS = (right oxygenized Hb (Hemoglobin) − left oxygenized Hb)/(right oxygenized Hb + left oxygenized Hb) [11].

### 2.2. Body Index

This research utilized a stabilometer to measure a body index. A Wii Balance Board was used, as it is known to have reliability comparable to medical equipment [11]. The purpose of using the Wii Balance Board was to measure the change in body balance after moisturizer application. Both left and right balance change was measured, and X axis data was taken as the X axis centered balance change. Experiment participants were asked to look at a tape of 3 cm as a marker, which was placed at 200 cm from the participant. The participants were asked to stand on a fixed point, with the inner sides of both feet placed on the designated area. The participants were asked to stand still for 40 s before the measurement, as it was necessary for them to be used to standing in such a position.

### 2.3. Subjective Index

The participants were asked to report their subjective feeling of balance to obtain the subjective index. When the participants were on the Wii Balance Board, they were asked to check how they felt about their balance or center of gravity, pre- and post-intervention (Figure 1).

### 2.4. Interventional Method

One lower leg was chosen to be the intervention side, which the participants reported with subjective measurement that they did not feel that they were putting their weight on. Moisturizer was applied to the lower leg on the intervention side. There have been several reports about the effect of touching in preceding studies, but the time of intervention varied from 10 to 20 min [12,13], and various parts of the body were touched. Since the intervention method was expected to be done as a part of daily nursing practice, the length of touching could not be too long. Therefore, the length of touching was set at 2 min. Unscented and inexpensive Vaseline was used as the moisturizer and is commonly used in medical care (White Vaseline; class 3 medical product; Kenei Seiyaku company, Fushimi-cho, Chuo-ku, Osaka-shi Osaka, Japan). Vaseline is used as a moisturizer that is known to protect the skin from external stimulus. It is widely used in medical practice for moisturization and the protection of wounds and is considered to be safe [14].

The direction of application was from under the knee to the backside of the lower leg. A preceding study reported that the parasympathetic nerve system was stimulated when a subject was touched at the speed of 3 to 5 cm per second, without applying strong force [3]. The therapist tried his best to maintain the touching speed of 3 to 5 cm per second and not to press too hard (see Figure 2 for the flow of the experiment).

### 2.5. Statistical Processing

The brain activity and heart rate were measured constantly by the HOT-1000 device from the beginning to the end of the entire process. Representative data were taken at the “Resting (pre)”, “After resting (post)”, “In the midst of moisturizer application (pre)”, and “After moisturizer application (post)” stages for 2 min each. “Resting” and “After resting” were considered “Non-Application”, and “In the midst of moisturizer application” and “After application” were considered “Application”. A two-way ANOVA was used to examine the main effects and the interaction. When a main effect was detected, the Holm method of multiple comparison was conducted.

Using the Wii Balance Board, X axis centered balance change was detected. The average value was obtained from data taken for 30 s. That average was considered the representative value. The data that were taken before intervention were interpreted as “minus” figures, and the data taken after the intervention were interpreted as “positive” figures, so that the figures taken pre- and post-intervention could be compared against each other. Three groups—“Before resting”, “After resting”, and “After moisturizing intervention”—were compared by one-way ANOVA. Subjective center of gravity was analyzed using a Chi-square test. A value of *p* < 0.05 was considered statistically significant. The statistical software employed was HAD [15].

## 3. Results

Ten healthy men and women with no physical ailments agreed to volunteer to participate in the experiment (five men and five women, average age 32.2 ± 37.9). Before the experiment, the participants were asked to feel their center of gravity subjectively. Six answered that they put more weight on the right leg, and four answered that they put more weight on the left leg. There was no one who answered that he/she put weight equally on both legs. Therefore, six participants received moisturizer application on their left legs, as they did not feel much weight on their left legs, and the other four received moisturizer application on their right legs, as they did not feel much weight on their right legs.

RLS did not demonstrate any statistical significance, either in the main effect or in the interaction (n.s.) (Table 1 and Table 2). For heart rate, the main effect was demonstrated in the group of post-moisturizer application (*p* < 0.01) (Table 3 and Table 4). The statistical significance was also examined using the Holm method (Table 5).

Concerning the X axis centered balance change, the weight shifted with statistical significance (*p* < 0.05) Table 6 and Table 7). Regarding the subjective balance, 9 out of 10 participants reported that they felt that they put more weight on the side of the leg which received the moisturizer application (*p* < 0.01) (Table 8, Table 9 and Table 10).

There was no significant difference between the main effect and the interaction. Also, the standard deviation was large and variation was observed. However, RLS decreased after skin care, therefore the Table 1 was presented as a positive result.

Main effect was observed, but no interaction was observed. In the multiple comparison, as shown in Table 2, significant difference was recognized. Especially in the midst of moisture application, heart rate decreased with statistical significance. After moisturizer application, heart rate was maintained at a low level. Therefore, moisturizer application may have contributed to the decrease in heart rate.

The center of gravity changed significantly with moisturizer application. Therefore, moisturizer application may have enhanced the sensitivity of the sole of the foot when firmly standing on the floor. Initial result of subjective center of gravity position

Before moisturizer application, four participants indicated “right”, six participants indicated “left”, and nobody indicated “the center”. After moisturizer application, nine out of ten participants said that their weight was more on the side where the moisturizer application was conducted. The subjective feeling changed after moisturizer application and the participants felt that their weight was cast more on the side to which the moisturizer was applied. The result matched with the objective measurement by a stabilometer. Therefore, moisturizer application may have improved the body balance of the participants.

## 4. Discussion

There have been a number of prior studies indicating a positive psychological effect of touching. One prior study on nurses at hospitals showed that they experience strong mental stress and sometimes burn-out. The study indicated that this was partly because of their demanding job and the fact that they did not have enough time to interact with the patients [16]. One nursing practice is to apply moisturizer to patients and give them massage. If nurses had enough time to give massages and interact with patients, it would be ideal. It may prevent nurses from experiencing severe stress and/or burn-out. However, another prior study found that the minimum massaging time to have any significant result would be 10 to 30 min. It would be impossible for nurses to give massage for 30 min to each in-patient. If a short period of moisturizer application could bring positive results, it would be very promising in helping nurses. Therefore, this study intended to examine whether moisturizer application to the lower legs for a short period of time would be effective. The psychological and physiological effect of moisturizer application for a short period time had never been studied before. This is the unique point of this research. In the geriatric care unit of hospitals, nurses apply moisturizer to the lower legs of the inpatients. Such practice is done every day, as the physiological functions of the skin of old people are compromised and moisturizer application needs to be done on a routine basis. If an optimal way to apply moisturizer to those inpatients was found, it would help not only the elderly people in the hospitals, but also the nurses, as they would have more time to interact with the patients.

Touching should not be done in a mechanical way. It should be done with respect and care and in a reciprocal manner. A prior study reported that massage done at a speed of 5 cm per second was most effective [3]. In this research, moisturizer application was done for 2 min at a speed of 5 cm per second. As physiological indices, RLS and heart rate were measured. As a body index, the body balance was measured. As a subjective index, subjective report of the center of gravity was measured.

Konno and Yoshikawa [17] reported that when their experiment participants found that their legs felt relaxed, they were more firmly putting their feet on the ground. In agreement with the work of Konno et al., the overall result of this research suggested that the experiment participants were relaxed after receiving moisturizer application on their lower legs, as their heart rate decreased, their X axis centered weight balance changed, and weight was more strongly felt on the side of the leg that received intervention. There was a good match between the results for weight balance and the subjective feeling of center of gravity. The results showed that the objective and the subjective outcomes matched.

In this research, the participants were asked to report their subjective feelings about their legs and the feeling of weight balance as one of the outcomes of the research. It naturally encouraged them to pay attention to their somatic awareness, which is a method often used in mindfulness practice and other psychological therapies. It is possible that the participants were naturally guided to pay attention to their somatic awareness in this research by being asked the questions about subjective feelings, and that this had a positive outcome. Seeing this result, one could claim that nurses in hospitals should be encouraged to guide their patients to pay attention to their legs when they apply moisturizer to their lower legs. It is clear from this research that skin moisturization by nurses can bring relaxation to patients, and if the patients are naturally guided to pay attention to their somatic sensations, it may enhance their relaxation, which may support their recovery.

The method used in this research was very simple. Anybody would be able to use it without special training. Therefore, it could be used by not only nurses but also by people in general. People could use the method on their family members and friends. Isolation and loneliness is a serious problem for elderly people. If the simple technique studied in this research was used to provide care for elderly people, it would help reduce the negative effects of isolation and loneliness. In this research, moisturizer application may have enhanced body balance. More research will be needed to examine the effect of this technique on body balance. If a positive result is confirmed, eventually this technique could be applied to elderly people to enhance their ability to balance. It could be used to prevent falls in elderly people.

## 5. Limitations

Some positive effects of moisturizer application were observed in this study. However, the sample size was small, and the number of experiment participants was only 10. In the future, it will be necessary to have more participants to build more reliable data.

Additionally, there were several other limitations to this research. First, there was no study of the psychological effect of different types of moisturizers or comparison of their functions and quality. In this research Vaseline was used. It would be necessary in the future to examine different types of moisturizers and their effects. Second, the functions of the prefrontal area are varied, and there are still many unknown factors about its nature. In this research, RLS was measured as an index of relaxation. However, more studies are required to assess the relaxation level of the participants and the state of the prefrontal area. Furthermore, psychological scales would need to be introduced so that relaxation state would be examined using various indices. Third, the research room conditions were quite different from those in the hospitals. Nurses would be busy and might be very much preoccupied. On the contrary, in this experiment, the therapist was relaxed, and there were no other people in the room, and it was quiet. A preceding study reported that the effect of touching changed according to the state of the therapist, even if they used the same touching modality [18]. It would be ideal if nurses could have more time to spend on each of their nursing practices and feel more comfortable when they are working with their patients.

## 6. Conclusions

In this research, moisturizer application to the lower legs of the experiment participants had positive psychological and physiological effects. After the moisturizer application, the participants were more relaxed and better balanced, in both objective and subjective indices. The intervention time was only two minutes, which was shorter than what has been practiced in preceding studies. Although the intervention time was very short, moisturizer application on the lower legs for two minutes still demonstrated a positive effect on relaxation in the experiment participants.

## Figures and Tables

**Figure 1 behavsci-08-00091-f001:**
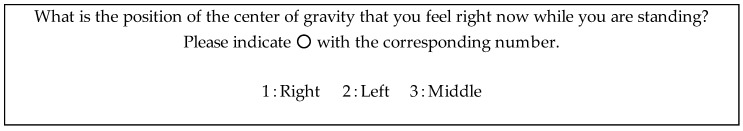
Interview method for subjective center of gravity position. Two measurements before and after the application intervention.

**Figure 2 behavsci-08-00091-f002:**
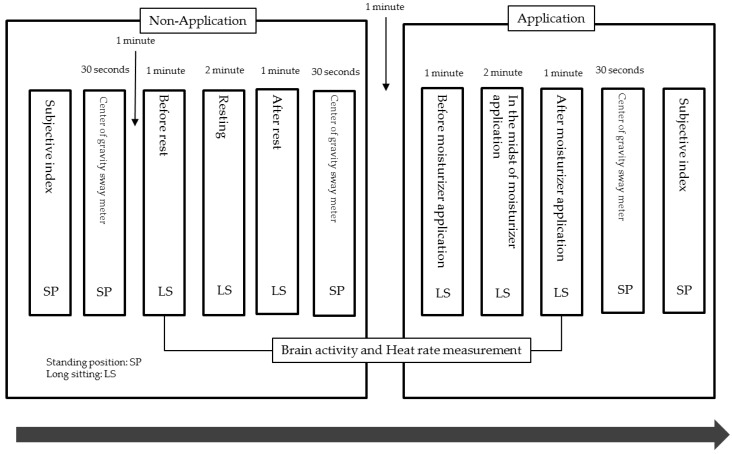
Experimental protocol (n = 10).

**Table 1 behavsci-08-00091-t001:** Average value and standard deviation of RLS (mmMm).

		Mean ± SD
Non Application	pre	−0.17 ± 1.08
	post	0.06 ± 0.87
Application	pre	0.42 ± 1.65
	post	−2.53 ± 8.55

**Table 2 behavsci-08-00091-t002:** Results of main effect and interaction by ANOVA (RLS).

Variable Name	*F* Value	Partial η^2^	df	*p* Value
Non Application-Application	0.92	0.09	9	0.36 *ns*
pre-post	0.83	0.08	9	0.39 *ns*
Interaction	1.08	0.11	9	0.33 *ns*

*ns*: not significant.

**Table 3 behavsci-08-00091-t003:** Average value and standard deviation of Heat rate (bpm).

		Mean ± SD
Non Application	pre	72.38 ± 11.24
	post	72.76 ± 10.49
Application	pre	66.39 ± 8.84
	post	67.25 ± 9.35

**Table 4 behavsci-08-00091-t004:** Results of main effect and interaction by ANOVA (Heat rate).

Variable Name	*F* Value	Partial η^2^	df	*p* Value
Non Application-Application	18.30	0.67	9	0.00 **
pre-post	1.42	0.14	9	0.26 *ns*
Interaction	0.24	0.03	9	0.64 *ns*

** *p* < 0.01, *ns*: not significant.

**Table 5 behavsci-08-00091-t005:** Multiple comparison analysis by Holm method (Heat rate).

Variable Name	Effect Size *d*	*t* Value	df	*p* Value
Resting—After rest	−0.03	−0.51	9	0.62 *ns*
Resting—In the midst of moisturizer application	0.57	3.42	9	0.02 *
Resting—After moisturizer application	0.47	3.48	9	0.03 *
After rest—In the midst of moisturizer application	0.63	4.51	9	0.01 *
After rest—After moisturizer application	0.53	5.47	9	0.00 **
In the midst of moisturizer application—After moisturizer application	−0.09	−1.28	9	0.23 *ns*

** *p* < 0.01, * *p* < 0.05, *ns*: not significant.

**Table 6 behavsci-08-00091-t006:** Average value and standard deviation of Center of gravity sway (cm).

	Mean ± SD
Before rest	−0.32 ± 0.38
After rest	−0.25 ± 0.34
After moisturizer application	0.27 ± 0.62

**Table 7 behavsci-08-00091-t007:** Multiple comparison analysis by Holm method (Center of gravity sway).

Variable Name	Effect Size *d*	*t* Value	df	*p* Value
Before rest	0.19	−0.67	9	0.52 *ns*
After rest	0.11	−3.02	9	0.03 *
After moisturizer application	1.00	−3.04	9	0.04 *

** *p* < 0.01, * *p* < 0.05, *ns*: not significant.

**Table 8 behavsci-08-00091-t008:** Initial result of Subjective center of gravity position.

Initial Result			
Appearance value	Frequency	Probability(%)	Application side
Right	4	40.00	Left
Left	6	60.00	Right
Middle	0	0.00	-
Total	10	100	

**Table 9 behavsci-08-00091-t009:** Initial result and change of Subjective center of gravity position after application.

Subjective Center of Gravity Position after Intervention		
Appearance value	Frequency	Probability(%)
Application side	9	90.00
Non-Application side	1	10.00
Middle	0	0.00
Total	10	100

**Table 10 behavsci-08-00091-t010:** Results of Subjective center of gravity position by Uniformity Test.

Uniformity Test	
Chi-squared	6.40
Degree of freedom	1
*p*-value	0.01 **

** *p* < 0.01.

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
