# Peer review of "An Experiment on the Psychological and Physiological Effects of Skin Moisturization on Lower Legs—In Expectation of Application to Nursing Practice at Hospitals"

_behavsci, 2018, doi:10.3390/bs8100091_

Round 1
Reviewer 1 Report
There are two comments as followings:
1. Line 115 &126 (figure 2). Please check the intervention time. Is it 2-minute or 3-minute (1 minute+ 2 minutes from the application stage at figure 2)
2. Line141. Please check p value. p>0.05 or p< 0.05?
Author Response
Dear Reviewer 1 and whom it may concern,
Thank you very much for your comments. Based upon your comments, I made revisions.
Taichi Hitomi
J.F. Oberlin University
Bunkyo University
Correspondence: t-hitomi@bgu.ac.jp;
Phone: +81-3-3814-1661

Reviewer 2 Report
This manuscript describes the study of the psychological and physiological effect of skin moisturization on lower legs. The parameters that were measured included brain activities, heart rate, and center of gravity unrest. The work is interesting, however, there are a few points which need to be addressed by the authors:
1) The presentation of the results are very brief. The authors may wish to include more details and description of the Tables that are provided.
2) More extensive discussion should be added to compare and evaluate the results with other similar reports. The purpose of the discussion is to interpret and describe the significance of your findings in light of what was already known about the research problem being investigated, and to explain any new understanding or insights about the problem after you've taken the findings into consideration.
3) The participants were 10 healthy male and female. Ten participants is considered as an insufficient sample. The authors should discuss that this small group cannot provide clear evidence. Presumably, more participants need to be evaluated.
Author Response
Dear Reviewer 2 and whom it may concern,
Thank you very much for your comments. I learned from your comment that my description of novelty of my research, which had not been studied in prior studies, was not sufficient. I did not mention that the sample size was small so that this small sample group would not provide clear evidence. Based upon your comments, I made revisions.
Taichi Hitomi
J.F. Oberlin University
Bunkyo University
Correspondence: t-hitomi@bgu.ac.jp;
Phone: +81-3-3814-1661

Round 2
Reviewer 2 Report
The manuscript was significant improved